# Investigating post-COVID-19 confidence in emergency use authorization vaccines: A hypothetical case of mpox

**Li Ping Wong[1,2,3]\*, Haridah Alias[2], Hai Yen Lee[4], Qinjian Zhao[5], Zhiwen Huang[2], Zhijian Hu[1], Yulan Lin[1]\***

**1** Department of Epidemiology and Health Statistics, Fujian Provincial Key Laboratory of Environmental Factors and Cancer, School of Public Health, Fujian Medical University, Fuzhou, Fujian, China, **2** Center for Population Health (CePH), Department of Social and Preventive Medicine, Faculty of Medicine, Universiti Malaya, Kuala Lumpur, Malaysia, **3** Department of Medicine, College of Medicine, Korea University, Seoul, Republic of Korea, **4** Tropical Infectious Diseases Research and Education Centre, Universiti Malaya, Kuala Lumpur, Malaysia, **5** College of Pharmacy, Chongqing Medical University, Chongqing, China

\* yulanlin@fjmu.edu.cn (YL); wonglp@ummc.edu.my (LPW)

## Abstract

### Background

The COVID-19 pandemic has profoundly influenced public trust in vaccines, particularly those authorized under Emergency Use Authorization (EUA). In light of the recent monkeypox (mpox) outbreak, the primary aim of this study is to uncover how experiences with the COVID-19 vaccination have shaped public trust in hypothetical EUA vaccines for mpox.

### Method

A nationwide cross-sectional survey using an online questionnaire was conducted across six regions in China. Trust in future EUA vaccines, influenced by the COVID-19 vaccination experience as well as other attitudinal and behavioral factors associated with the willingness to receive an EUA-authorized mpox vaccine, was analyzed using partial least squares structural equation modelling (PLS-SEM).

### Results

The overall willingness to receive an EUA-authorized mpox vaccine across all regions is 67.9%. The less economically developed Northeastern and Northwestern regions, with willingness rates of 84.9% and 81.8%, respectively, showed higher intention to be vaccinated than the more affluent Eastern region, which reported a willingness rate of 71.2%. There is a generally positive influence of the COVID-19 vaccination experiences on trust in future EUA vaccines, with a combined total of 61.8% (41.5% reported a somewhat increased level of trust and 20.3% reported a

**Data availability statement:** All data files are available from the Zenodo: https://zenodo.org/records/14195496.

**Funding:** The author(s) received no specific funding for this work.

**Competing interests:** The authors have declared that no competing interests exist.

significantly increased level of trust). This trust emerged as the strongest determinant of vaccination willingness ($\beta = 0.441$, $p < 0.001$). Other factors influencing willingness include fear of the disease ($\beta = 0.096$, $p < 0.001$), knowledge ($\beta = 0.064$, $p = 0.017$), and prevention practices ($\beta = 0.066$, $p = 0.005$). Additionally, significant but smaller effects were noted for education ($\beta = 0.059$, $p = 0.005$) and urban-rural locality ($\beta = -0.053$, $p = 0.012$).

## Conclusion

Findings underscore the importance of bolstering public trust in EUA vaccines, as well as addressing related attitudinal and behavioral factors, to enhance acceptance of mpox vaccines. This requires transparent communication about the rigorous approval process, continuous safety monitoring, and real-world effectiveness of EUA vaccines.

### Author summary

The COVID-19 pandemic has shaped how people view vaccines, especially those approved quickly for emergencies. This study explores how these experiences affect trust in a potential mpox vaccine in China. A nationwide survey of 2,000 people found that about 68% were willing to get an mpox vaccine if one became available. Willingness was higher in less wealthy regions, like the Northeast and Northwest, compared to richer areas. Trust in emergency-approved vaccines, influenced by COVID-19 vaccination experiences, was the biggest factor driving people's willingness to get vaccinated. Fear of mpox and taking preventive measures, like handwashing and avoiding close contact, also played a role. While many people said their trust in emergency vaccines increased after COVID-19, some still worried about how quickly these vaccines are approved and their long-term safety. The study suggests that clear communication, addressing concerns, and ensuring fair access to vaccines are key to building trust and acceptance for future health emergencies like mpox.

## Introduction

The COVID-19 pandemic has brought global attention to the importance of vaccines in preventing and controlling infectious diseases. While the unprecedented speed of development and the successful rollout of the Emergency Use Authorization (EUA) COVID-19 vaccines has been applauded as a public health success story, it also spurred considerable controversy and hesitancy even amongst individuals who did not previously hold anti-vaccination stances [1]. During the COVID-19 pandemic, the remarkably rapid development of the COVID-19 vaccine and its widespread global distribution have undoubtedly influenced public confidence in vaccines [2]. This influence has been observed both in positive and negative ways.

On the positive side, the successful rollout and containment of the pandemic through vaccination efforts may have increased people's confidence in vaccines [3]. On the negative side, instances of unexpected side effects or adverse reactions to the COVID-19 vaccine have led to hesitancy and skepticism among some individuals, impacting their overall trust in vaccines [4,5]. According to a study in the UK, there has been a notable decline in vaccine confidence since the beginning of the pandemic [1]. As of today, there have been limited studies on vaccine confidence for future pandemics following the COVID-19 pandemic.

Monkeypox (mpox) is a viral disease caused by the monkeypox virus, part of the same family of viruses as smallpox. It was first discovered in 1958 in laboratory monkeys, with the first human case recorded in 1970 in the Democratic Republic of Congo. The disease primarily occurs in Central and West Africa, often in proximity to tropical rainforests, and is typically transmitted to humans from animals such as primates. Around the year 2022, mpox began spreading more widely outside endemic regions, raising global concern [6]. In 2022, the WHO declared mpox a Public Health Emergency of International Concern (PHEIC) marking its spread beyond traditionally endemic regions in Central and West Africa to over 75 countries [7]. Despite containment efforts, the WHO declared a second emergency on 14th August 2024 due to continued spread [8] following a significant global outbreak. During the first PHEIC, China reported only one imported case in September 2022 [9]. However, by December 31, 2023, the number of confirmed cases had risen to 1,712, spanning 29 provincial-level administrative divisions (PLADs) [10]. The 39th situation report for the multi-country outbreak of mpox, covering the period from January 1, 2022, to June 30, 2024, found that China had a total of 2,460 confirmed cases and one death [11].

Similar to the COVID-19 pandemic, scientists worldwide have rapidly mobilized to develop a vaccine against mpox, recognizing that vaccination is the most effective tool to curb the spread of the virus [8]. Given these global efforts, it is increasingly important to study public confidence in a hypothetical mpox vaccine post-COVID-19 pandemic. Understanding how the COVID-19 pandemic has influenced vaccine confidence is essential for shaping effective public health responses, as it can impact vaccine uptake and the success of future vaccination initiatives. Therefore, examining the level of acceptance and trust in a mpox vaccine, especially in light of lessons learned from the COVID-19 pandemic, is critical. It is important to note that while existing vaccines used to protect against mpox provide some level of immunity, their limitations should be acknowledged. JYNNEOS, a non-replicating, live attenuated vaccine, is FDA-approved for the prevention of both smallpox and mpox [12]. However, JYNNEOS was originally developed as a smallpox vaccine with potential cross-protection against other orthopoxviruses rather than being explicitly designed for mpox [13]. Therefore, the development of vaccines specifically designed for mpox, along with vaccine acceptance, remains a key priority in addressing its emergence as an infectious disease.

The rise of mpox has heightened concerns about public health preparedness in China. Therefore, this study has several important aims. The primary aim of this study is to uncover how experiences with the COVID-19 vaccination have shaped public trust in future EUA vaccines for infectious disease pandemics, with a particular emphasis on the ongoing mpox outbreak. Due to the novelty of the mpox pandemic, we also aim to assess the general public's current level of knowledge about mpox symptoms and their preventive practices related to mpox infection, following the declaration of mpox as a PHEIC. Specifically, we examine preventive behaviors such as avoiding close contact, practicing good hygiene, avoiding face-touching, using personal protective equipment (PPE), and staying informed by keeping updated with public health guidelines and advice from health authorities regarding mpox [14,15], in the event that an EUA vaccine for mpox is developed and made available during an emergency, our goal is to evaluate public attitudes toward EUA vaccines. Additionally, this study also examines the public's willingness to receive an EUA mpox vaccine and their willingness to pay (WTP) for it if it becomes available.

This research could provide valuable insights for policymakers and health authorities in China, helping to design better communication strategies, address concerns proactively, and ensure higher vaccination rates in the event of future pandemics involving emerging neglected tropical diseases (NTDs) like mpox in China. By gaining a clear understanding of the population's views, concerns, and preferences, health authorities can tailor effective communication campaigns, address

potential barriers, and design vaccination programs that maximize acceptance and uptake of the vaccine. Ultimately, findings will contribute to better preparedness and management of mpox, should it ever reach a pandemic level in China.

## Methodology

### Ethics statement

This study adhered to the principles of the Declaration of Helsinki and received ethical approval from the Fujian Medical University Research Ethics Committee, China (Approval: FJMU 2024 NO.300). Participants were informed about the voluntary nature of their involvement, and they could withdraw from the survey at any time without providing justification. Written consent to participate was obtained through a consent statement displayed on the first page of the online survey, where participants were required to click an agreement button to proceed. The ethics committee approved the online consent process, and no identifying information was collected from participants during the survey. Consent for publication was also obtained, ensuring participants were aware that their data would be used for research and publication purposes. All data were handled confidentially and stored securely in compliance with data protection regulations. Participants did not receive any financial incentives for their participation in the study. There were no anticipated risks or potential harms associated with taking part in the research.

### Study participants and survey design

A nationwide cross-sectional survey using an online questionnaire was conducted from 25th August to 26th September 2024. The inclusion criteria were Chinese citizens above the age of 18 years and able to comprehend and read Chinese. A convenience sampling method was employed, with a survey company distributing the survey link through its existing database. Participants received the survey link via their mobile WeChat accounts. Participants were collected nationwide across six regions in China. The sample size was calculated using the formula: $n = Z^2 P(1-P)/d^2$. Using a margin of error of 0.05 (5%), with a 95% CI and 50% response distribution, the calculated sample size was 384 for each region. The sample size was multiplied by the predicted design effect of two to account for the use of convenience sampling and an online survey [16]. Hence, the optimal sample size was set to 768 (384 x 2).

### Instruments

The questionnaire (S1 Appendix) consisted of eight sections: 1) demographic information, 2) knowledge of mpox symptoms, 3) risk perception regarding mpox, 4) preventive practices, 5) attitudes towards vaccines authorized under EUA, 6) trust in future EUA vaccines influenced by the COVID-19 vaccination, and 7) willingness to receive an EUA-authorized mpox vaccine and willingness to pay for it.

**Demographics background.** Personal details collected include age, gender, highest educational level, monthly household income, locality, and current residing region.

**Knowledge about mpox symptoms.** The knowledge assessment comprised 8 items evaluating symptoms of mpox, including rash, fever, chills, swollen lymph nodes, fatigue, muscle aches, headaches, and respiratory symptoms. The response options were "true", "false", and "don't know". A score of 1 was awarded for correct answers, while both "false" and "don't know" responses received a score of 0. The total possible score ranges from 0 to 8, with higher scores indicating a greater level of knowledge.

**Risk perception towards mpox.** Perceptions of mpox were assessed through questions on perceived susceptibility, severity, and fear of the disease following the WHO's declaration of mpox as a PHEIC. Responses were recorded on a 4-point Likert scale, ranging from "not at all," "slightly," "moderately," to "very."

**Prevention practices.** To assess preventive behaviors against mpox, participants were asked five questions related to avoiding close contact with people who have a rash or are suspected of being infected with mpox, practicing good

hygiene, avoiding face-touching, using personal protective equipment (PPE), and staying informed. Responses were measured using a 4-point Likert scale: always, often, sometimes, and never. Each response was scored from 0 to 3, with "always" receiving 3 points, "Often" receiving 2 points, "sometimes" receiving 1 point, and "never" receiving 0 points. The total score for preventive behaviors ranged from 0 to 15, with higher scores reflecting greater adherence to preventive measures against mpox.

**Attitudes towards EUA vaccine.** Participants' perceptions of vaccines approved under EUA were assessed through 10 statements addressing concerns about the approval process, quality control, effectiveness, transparency, long-term side effects, and the potential impact on public trust. Responses were scored on a 4-point Likert scale, where "strongly agree" was scored as 1 and "strongly disagree" as 4. The total score ranged from 10 to 40, with higher scores reflecting more positive perceptions and trust in EUA vaccines.

**Trust in future EUA vaccines influenced by the COVID-19 vaccination.** This section assessed lessons learned from EUA COVID-19 vaccination on the confidence in the hypothetical EUA vaccine for mpox. To evaluate the impact of the EUA COVID-19 vaccination on trust in hypothetical EUA vaccines for mpox, participants were asked a single question: "To what extent has the EUA COVID-19 vaccination affected your trust in a future EUA vaccine for infectious disease pandemics such as mpox?" Responses were recorded using a 5-point Likert scale, with options ranging from "My trust has significantly decreased" to "My trust has significantly increased".

**Vaccination willingness and WTP.** To assess willingness to be vaccinated with an EUA vaccine for mpox if it was developed and authorized for emergency use, participants were asked two questions. The first question inquired if they would be willing to receive a vaccine for mpox if developed and authorized for emergency use, with responses ranging from "not willing" to "extremely willing" on a 5-point Likert scale. The second question explored preferences regarding the type of vaccine, asking if participants would be willing to receive an mRNA vaccine, a traditional vaccine, either type, or none at all. A note was provided to participants for clarification: "an mRNA vaccine is a type of vaccine that uses messenger RNA to instruct cells in the body to produce a protein that triggers an immune response. This response helps the body recognize and fight the actual virus if it encounters it in the future. On the other hand, a conventional (traditional) vaccine typically uses a weakened or inactivated form of the virus, or a protein from the virus, to stimulate the immune system to recognize and combat the virus without causing the disease" This evaluation aimed to gauge public willingness to accept potential mpox vaccines and their preferences for vaccine types. The final question assesses the maximum amount participants are willing to pay for a dose of the mpox vaccine, should it become available.

## Statistical analysis

The sample characteristics were summarized using means ± standard deviations (SD) or frequencies, depending on the variable types. The internal consistency of the items was assessed using Cronbach's alpha coefficient for the relevant domains. The Cronbach's α values were 0.849 for the knowledge score, 0.811 for preventive practices, and 0.886 for attitudes towards the EUA vaccine, indicating good reliability; an alpha coefficient between 0.6 and 0.7 suggests an acceptable level of reliability, while a coefficient of 0.8 or higher indicates very good reliability [17]. Partial least squares structural equation modeling (PLS-SEM) was employed to identify factors influencing vaccination willingness. This analytic method effectively evaluates the reliability of the dataset, the statistical significance of coefficients, and the errors in the estimated path coefficients. Bootstrapping was conducted using Smart PLS software version 3.2.8 (SmartPLS GmbH). All statistical analyses were performed using the IBM SPSS Version 20.0 (IBM Corp, Armonk, NY, USA) [18].

## Results

A total of 2000 participants completed responses were received and included in the analysis, with a majority aged between 26 and 35 years (49.5%) and 29.9% aged 18–25. The gender distribution revealed a higher proportion of males (62.6%) compared to females (37.5%). In terms of education, half of the participants (50.0%) held a bachelor's degree, while 11.1%

had a secondary school education or less. Regarding annual family income, as of 2022, the average disposable income per capita in China was approximately CNY 36,883 annually [19]. Assuming an average household size of about 3 people (based on recent demographic data) [20], the average annual household income would be CNY 110,649. In our study, an approximately equal proportion of participants reported annual family incomes in the CNY 100,001–150,000 range (32.9%) and the > 150,000 CNY range (28.5%), reflecting a relatively balanced distribution of middle-to-upper-income households. The majority of participants resided in urban areas (57.6%), with 29.2% from the eastern region of China, followed by 25.1% from the southern central region. The demographic characteristics of the study participants reflect a predominantly young, educated, and urban population with relatively high-income levels (Table 1). Compared to participants in the undecided/somewhat unwilling/unwilling groups, those in the extremely willing/somewhat willing groups were more likely to have a higher education level, a higher annual family income, and reside in urban areas (all p < 0.05).

**Table 1. Demographic characteristics of study participants and willingness to receive an EUA mpox vaccine by demographic characteristics (N = 2000).**

| Characteristics | N (%) | Willingness to receive an EUA Mpox vaccine | | |
| --- | --- | --- | --- | --- |
| | | Extremely willing/ Somewhat willing (n = 1359) | Undecided/ Somewhat unwilling/ Unwilling (n = 641) | p-value |
| Age group (years) | | | | |
| 18-25 | 597 (29.9) | 389 (65.2) | 208 (34.8) | 0.123 |
| 26-35 | 990 (49.5) | 693 (70.0) | 297 (30.0) | |
| 36-60 | 413 (20.7) | 277 (67.1) | 136 (32.9) | |
| Gender | | | | |
| Male | 1251 (62.6) | 868 (69.4) | 383 (30.6) | 0.083 |
| Female | 749 (37.50 | 491 (65.6) | 258 (34.4) | |
| Highest education level | | | | |
| Secondary school and below | 221 (11.1) | 133 (60.2) | 88 (39.8) | <0.001 |
| High school/technical school | 663 (33.2) | 395 (59.6) | 268 (40.4) | |
| Bachelor | 999 (50.0) | 749 (75.0) | 250 (25.0) | |
| Postgraduate | 117 (5.9) | 82 (70.1) | 35 (29.9) | |
| Annual family income (CNY) | | | | |
| <50,000 | 270 (13.5) | 167 (61.9) | 103 (38.1) | 0.030 |
| 50,000- 100,000 | 504 (25.2) | 336 (66.7) | 168 (33.3) | |
| 100,001-150,000 | 658 (32.9) | 448 (68.1) | 210 (31.9) | |
| >150,000 | 568 (28.5) | 408 (71.8) | 160 (28.2) | |
| Birthplace | | | | |
| Urban | 1152 (57.6) | 867 (75.3) | 285 (24.7) | <0.001 |
| Sub-urban | 452 (22.6) | 262 (58.0) | 190 (42.0) | |
| Rural | 396 (19.8) | 230 (58.1) | 166 (41.9) | |
| Region | | | | |
| Eastern | 584 (29.2) | 416 (71.2) | 168 (28.8) | <0.001 |
| Northeastern | 271 (13.6) | 230 (84.9) | 41 (15.1) | |
| Northern | 440 (22.0) | 280 (63.6) | 160 (36.4) | |
| Northwestern | 77 (3.9) | 63 (81.8) | 14 (18.2) | |
| Southern Central | 502 (25.1) | 297 (59.2) | 205 (40.8) | |
| Southwestern | 126 (6.3) | 73 (57.9) | 53 (42.1) | |

†1 CNY = 0.138 USD.

The proportion of correct responses regarding knowledge of mpox symptoms is shown in S2 Appendix. Fever was recognized as a symptom of mpox by 50.4% of respondents, while rashes were identified by 49.8%. Muscle aches and backaches were recognized by 47.9%, and exhaustion was correctly identified by 47.3%. Respiratory symptoms were noted by 47.2% of participants, while headache was identified by 44.5%. Chills were correctly noted by 41.9%, and swollen lymph nodes were recognized by 39.5% of respondents. The total knowledge score ranged from 0 to 8, with a mean of 3.6 (SD ± 2.9).

As shown in S3 Appendix, in terms of perceived susceptibility, the majority (52.2%) believed they were "somewhat likely" to contract mpox, while 27.4% felt it was "not very likely," and only 14.0% considered it "very likely." For perceived severity, nearly half (44.9%) of the participants considered mpox to be "moderately severe" if contracted, and 25.7% regarded it as "very severe." In the assessment of fear of mpox, 48.8% were "somewhat concerned" about its risks, and 26.8% were "very concerned". Overall, the findings suggest that participants acknowledged a moderate likelihood of contracting mpox and were generally concerned about its severity and potential consequences.

S4 Appendix showed that a large portion reported "often" or "always" avoiding close physical contact with individuals suspected of having mpox (44.9% and 26.7%, respectively). Similarly, 42.5% washed their hands frequently, and 37.8% always did so. Regarding avoiding face-touching with unwashed hands, 40.4% followed this practice often, while 30.2% always adhered to it. In crowded places, 40.2% often wore masks, and 32.1% always did so. Lastly, staying updated with public health guidelines was common, with 41.2% doing so often and 31.4% always. A small percentage of participants (ranging 5.2 to 9.9%) reported "never" engaging in these practices. The total preventive practices score ranged from 0 to 15, with a mean value of 9.9 (SD ± 3.3).

S5 Appendix reveals mixed attitudes towards EUA vaccines with a considerable proportion expressing concerns about the rapid approval process (57.0% agreed, 25.5% strongly agreed) and quality control (47.6% agreed, 32.7% strongly agreed). Many participants also worried about the effectiveness of EUA vaccines compared to fully approved ones (46.7% agreed, 29.5% strongly agreed) and potential unknown long-term side effects (48.8% agreed, 27.6% strongly agreed). Despite these concerns, a majority acknowledged the necessity of EUA vaccines during pandemics (51.7% agreed, 27.5% strongly agreed), with 53.1% willing to receive an EUA vaccine if it were the only option. Additionally, 48.1% believed the benefits outweigh the potential risks, though 49.2% feared that the use of EUA vaccines might undermine public trust in vaccines overall. The total attitudes towards EUA vaccines ranged from 10 to 36, with a mean value of 23.2 (SD ± 3.63).

Finding on single item assessing how trust in future EUA vaccines influenced by the COVID-19 vaccination experience impacted the level of trust in future EUA vaccines revealed that a total of 2.8% reported that their trust significantly decreased, while 7.6% indicated a somewhat decreased level of trust. In contrast, 27.8% felt their trust remained the same. A notable portion of participants reported an increase in trust, with 41.5% stating it somewhat increased and 20.3% indicating it significantly increased. Overall, these suggest a generally positive influence of the EUA COVID-19 vaccination on trust in future EUA vaccines, with a combined total of 61.8% of participants (41.5% somewhat increased + 20.3% significantly increased) expressing an increased level of trust.

Fig 1 shows the proportions of individuals who are somewhat or extremely willing to receive an EUA-authorized mpox vaccine across different regions in China. Overall, 27.9% (n = 557) expressed being extremely willing, while 45.6% (n = 912) reported being somewhat willing to receive the vaccine. The highest willingness was observed in the Northeastern region, with 84.9% (n = 230) expressing willingness to receive the vaccine, followed by the Northwestern region at 81.8% (n = 63). In the Eastern region, 71.2% (n = 416) were willing, while the Northern region showed 63.6% (n = 280). The willingness was lower in the Southern and Southwestern regions, with 59.2% (n = 297) and 57.9% (n = 73), respectively. These findings indicate regional variation in vaccine acceptance, with generally higher rates in the northeastern and northwestern areas. The overall willingness to receive the mpox vaccine across the regions in China is approximately 67.9%. Willingness to receive the mpox vaccine by demographics is shown in Table 1, with the highest willingness observed

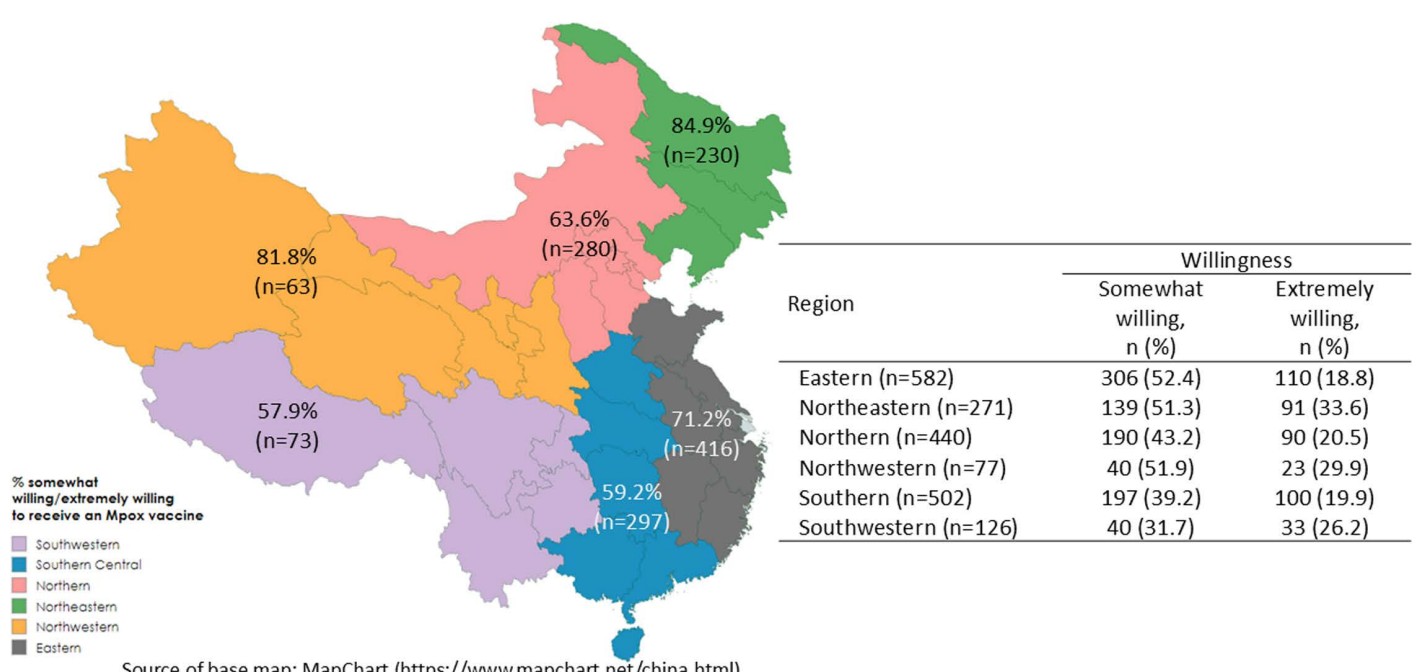

**Fig 1. Proportion of willingness to receive vaccine for Mpox by region.** Created with mapchart.net (https://www.mapchart.net/china.html).

| Region | Willingness | |
|---|---|---|
| | Somewhat willing, n (%) | Extremely willing, n (%) |
| Eastern (n=582) | 306 (52.4) | 110 (18.8) |
| Northeastern (n=271) | 139 (51.3) | 91 (33.6) |
| Northern (n=440) | 190 (43.2) | 90 (20.5) |
| Northwestern (n=77) | 40 (51.9) | 23 (29.9) |
| Southern (n=502) | 197 (39.2) | 100 (19.9) |
| Southwestern (n=126) | 40 (31.7) | 33 (26.2) |

among those aged 26–35 (70.0%), though age and gender differences were not statistically significant. Education level played a key role (p<0.001), with higher willingness among those with a bachelor's (75.0%) or postgraduate degree (70.1%) compared to lower education levels. Income also influenced willingness (p=0.030), with the highest acceptance among those earning >150,000 CNY (71.8%). Additionally, urban residents (75.3%) were significantly more willing than those from suburban (58.0%) and rural areas (58.1%) (p<0.001).

Fig 2 presents the PLS-SEM analysis of factors associated with the willingness to receive an mpox vaccine. The strongest positive association with willingness is trust in EUA vaccines (β=0.441, p<0.001), followed by fear of the disease (β=0.096, p<0.001), and prevention practices (β=0.064, p=0.017). Knowledge about mpox was also negatively associated with vaccination willingness; however, the association was not significant (β=-0.007, p=0.715). Other significant but smaller positive effects include education (β=0.059, p=0.005) and susceptibility to the disease (0.066, p=0.005). Negative associations were found with birthplace location (β=-0.053, p=0.012), whereby participants from urban areas exhibited higher willingness. Income showed a non-significant negative association (β=-0.022, p=0.299) with willingness. The R-squared adjusted for the model is 0.290, indicating that the included factors explain 29% of the variance in willingness to receive an mpox vaccine. Gender, age, and attitude towards EUA vaccines had no significant impact on willingness to receive the vaccine.

Descriptive analysis of vaccine preference shows that a minority, 12.7% (n=254), prefer an mRNA vaccine, while 38.3% (n=767) favor a traditional conventional vaccine, and 34.0% (n=679) indicated they would accept either type of vaccine. Meanwhile, 6.4% (n=128) stated they did not want to receive any mpox vaccine, and 8.6% (n=172) were unsure about the difference between mRNA and traditional vaccines.

The final question assesses the maximum amount participants are willing to pay for a dose of the mpox vaccine, should it become available. Results showed that the majority of participants (79.4%) did not specify a maximum price. Of those who did, the most common amount was CNY 100 (3.2%), followed by CNY 200 (2.8%) and CNY 50 (1.8%). Smaller

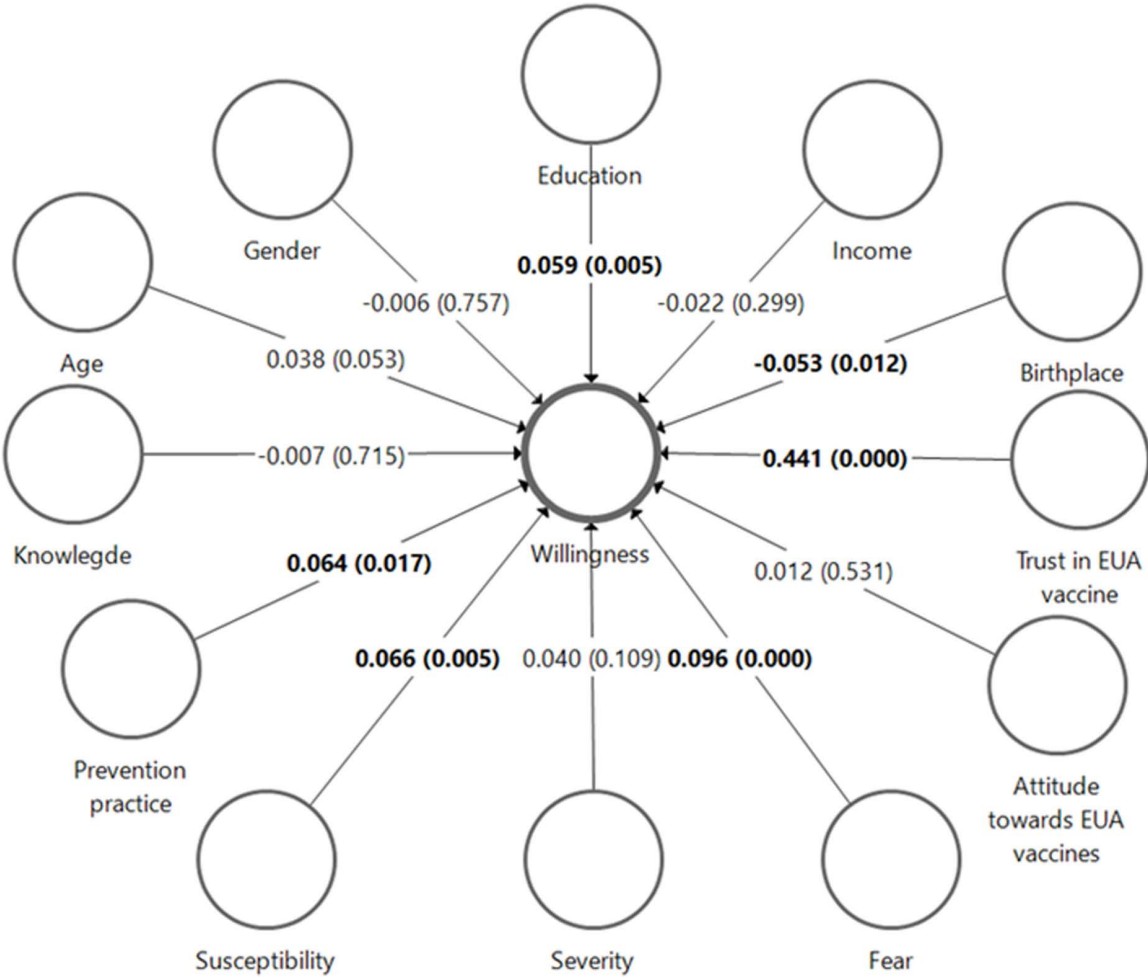

**Fig 2. PLS-SEM for factors associated with willingness to receive an Mpox vaccine.**

portions were willing to pay amounts like CNY 150 (1.3%) and CNY 300 (0.8%). A few participants listed lower or higher amounts, such as CNY 10 (0.3%) or CNY 500 (0.8%), with a few outliers.

## Discussion

The COVID-19 pandemic and the rapid development of vaccines for emergency use have significantly influenced public perceptions of Emergency Use Authorization (EUA) vaccines. This study explores the implications of vaccine confidence for future pandemics, focusing on a hypothetical scenario involving the mpox vaccine, particularly in the context of its recent designation as a Public Health Emergency of International Concern (PHEIC). This research is particularly timely, as the WHO has recently approved Bavarian Nordic's mpox vaccine, which currently targets adolescents aged 12–17, an age group considered especially vulnerable to mpox [21].

### Knowledge about mpox symptoms and risk perception

The study on mpox symptom knowledge, based on the weighted proportions of correct responses, indicates that participants had a moderate understanding overall. Fever and rashes were the most recognized symptoms, while awareness of

less common symptoms like swollen lymph nodes and chills was notably lower. Since no item on symptom was correctly identified by more than half of the participants, it suggests that the general knowledge of mpox symptoms is moderate. This points to a clear need for increased public education and awareness campaigns to cover the full range of mpox symptoms. The lack of knowledge may be due to the absence of reported mpox cases in China and the limited media coverage of the disease. It is recommended that public health authorities launch comprehensive educational campaigns to raise awareness of mpox, even though it is not currently a pandemic in China, to ensure public preparedness. The findings suggest that participants had a moderate perception of their susceptibility to mpox, likely due to the absence of reported mpox cases during the study period, which resulted in a lower sense of urgency and perceived risk. The absence of mpox cases and limited media coverage likely contributed to gaps in knowledge, emphasizing the need for targeted public health interventions to boost awareness and preparedness.

### Prevention practices

Findings on preventive measures, such as avoiding close contact, frequent handwashing, mask-wearing in crowded places, and staying updated with public health guidelines, showed that these practices were widely adopted, with the majority of participants often or always adhering to them. However, the total preventive practices score suggests that further efforts are needed to encourage consistent and widespread compliance with health recommendations. Similarly, the absence of reported mpox cases in China during the study period likely reduces the perceived urgency of maintaining strict preventive behaviors among the population. Strengthening public health messaging, even in the absence of a local outbreak, could help sustain a higher level of precautionary measures and ensure preparedness should the situation change [22].

### Attitudes towards EUA vaccine

The results on attitudes toward EUA vaccines among participants revealed that many expressed concerns about various aspects of these vaccines, particularly regarding the rapid approval process and quality control. Doubts about their effectiveness compared to fully approved vaccines were also prevalent. The belief that EUA vaccines are less effective than fully approved vaccines may arise from several factors. Firstly, EUA vaccines receive authorization based on interim clinical trial data rather than extensive long-term studies, raising concerns about their reliability [23,24]. Additionally, the EUA designation is often associated with an experimental or incomplete approval process. Past instances of expedited vaccine rollouts with unexpected side effects may further deepen skepticism [24,25]. Addressing these concerns through clear communication and transparency in vaccine development and authorization processes is crucial for maintaining public trust. Additionally, worries about potential unknown long-term side effects reflected broader societal skepticism toward rapidly developed vaccines during health emergencies, consistent with findings in other studies, which have highlighted similar hesitations among populations regarding new COVID-19 vaccines [26,27]. Despite these concerns, many participants acknowledged the necessity of EUA vaccines during pandemics and indicated a willingness to receive one if it were the only option available. This willingness aligns with previous research showing that, while apprehensions about safety exist, individuals often recognize the critical role of vaccines in controlling infectious diseases. On a positive note, a considerable portion of respondents believed that the benefits of EUA vaccines outweighed potential risks, illustrating a pragmatic perspective. However, a considerable number also feared that the use of EUA vaccines could undermine public trust in vaccines overall, echoing concerns raised in many reports that emphasize the importance of maintaining confidence in vaccine development, especially in the context of emergency use approvals with accelerated development timelines [28]. Therefore, public health strategies in China should inform and reassure the public about the rigorous standards that underpin vaccine development.

### Lessons learned from COVID-19 vaccination on trust in future EUA vaccines

Overall, the responses suggest that the success of EUA vaccines during the COVID-19 pandemic has positively impacted public trust in future EUA vaccines for infectious diseases. The positive shift in trust observed in this study may be linked

to China's unique vaccination approach during the COVID-19 pandemic. Unlike many countries that used mRNA vaccines, China primarily relied on inactivated virus vaccines like Sinovac and Sinopharm. These vaccines, based on traditional technology, may have fostered higher public confidence due to their established safety record. During the COVID-19 pandemic, China's decision to primarily use inactivated virus vaccines, instead of mRNA vaccines was influenced by a combination of scientific, logistical, and strategic factors. One key reason was the familiarity and established track record of inactivated vaccine technology, which had been used for decades in China and globally for diseases such as polio and Hepatitis A [29]. This established track record likely bolstered confidence in the safety and effectiveness of inactivated vaccines among regulators, manufacturers, and the general public Additionally, China prioritized domestic production capacity to ensure supply security and reduce dependence on foreign pharmaceutical companies [30]. Another factor was the challenges China faced in mRNA vaccine development during the early phase of the pandemic, including limited expertise and restricted access to specialized technologies.

This familiarity likely contributed to the increased trust in future EUA vaccines, including for mpox. Research shows that trust in vaccines is often influenced by the perceived safety and consistent public health messaging [31,32]. China's success with its COVID-19 vaccination campaign and the population's greater confidence in familiar vaccine platforms may explain the higher trust levels compared to countries that used newer vaccine technologies, where skepticism was more prevalent [33]. In China, the National Medical Products Administration (NMPA) has established clear guidelines for EUA vaccines to ensure that safety and efficacy. The guidelines for vaccines development in China require comprehensive preclinical studies, including laboratory and animal testing, to evaluate the vaccine's safety and immunogenicity. While clinical trials typically progress through Phases I, II, and III to evaluate a vaccine's safety and effectiveness, EUA allows for expedited approval based on interim data from early-phase trials, requiring at least Phase II data, if not Phase III. Even under accelerated timelines, these trials must meet stringent scientific and ethical standards. Vaccine manufacturers must adhere to good manufacturing practices to ensure the consistency and quality of vaccine production. Additionally, post-authorization monitoring is also a critical component, with robust pharmacovigilance systems in place to track adverse events and long-term effects after the vaccine is deployed [34].

Nevertheless, the country has since emphasized the importance of learning from the West's advancements in mRNA vaccine technology and has called for global collaboration with an open-minded approach [35]. Therefore, there is a need to enhance public trust in mRNA vaccine technology for future pandemics as well as in the development of vaccines for NTDs such as mpox, as the future of vaccine development is increasingly shifting toward mRNA due to its superiority over traditional vaccines.

### Vaccination willingness and WTP

The disparities in willingness to receive the mpox vaccine across different regions in China may be influenced by the varying economic profiles of these regions. Northeastern and Northwestern regions, which showed the highest willingness rates of 84.9% and 81.8%, are less economically developed compared to the more affluent Eastern region, which reported a willingness rate of 71.2%. In contrast, the economically developed Northern region exhibited a moderate willingness of 63.6%, suggesting that higher economic development does not necessarily translate to higher vaccine acceptance. The lower willingness rates in the Southern (59.2%) and Southwestern (57.9%) regions, which are known for significant economic disparity, further highlight the complex relationship between regional economic conditions and vaccine confidence. Further research is needed to understand the underlying reasons for these disparities. Regional disparities in vaccine acceptance in China remain uncertain in the literature. In the context of COVID-19 vaccine, some studies have reported higher COVID-19 vaccine acceptance in the more developed eastern region [36,37], while others have found lower acceptance in the same region [38,39]. Nonetheless, tailored communication strategies and outreach efforts could be vital in boosting vaccine confidence, particularly in regions where acceptance remains low. The overall willingness across all regions is approximately 67.9%, which, while moderately high, underscores a need for targeted public health interventions

in regions with lower acceptance. Similar strategies have been successfully employed during the COVID-19 vaccination campaign, where digital platforms, community and healthcare workers engagement, and transparent communication were used to address hesitancy and improve coverage [40]. Additionally, while vaccination rates in Chinese cities remain relatively high due to smooth mobilization and strong health risk awareness, rural areas have faced challenges in uptake. To address this, targeted efforts such as free shuttle services, extended vaccination hours, mobile healthcare teams, and door-to-door education campaigns have been implemented to improve coverage [41].

The PLS-SEM analysis highlights several factors associated with the willingness to an EUA-authorized mpox vaccine, with trust in EUA vaccines emerging as the strongest positive predictor. This is consistent with findings from previous studies in a systematic review, which highlighted the crucial role of trust in government-endorsed vaccines in shaping public acceptance, particularly during emergencies [42]. The significant influence of fear of the disease and perceived susceptibility is consistent with studies on COVID-19 vaccine acceptance, where risk perception drives vaccine uptake [43,44]. However perceived severity is not significant, this could be due to during the study period there are no mpox cases yet in China.

The finding that prevention practices about mpox have also been identified as important in shaping health behaviors, echoing earlier research emphasizing the role of awareness and proactive measures in COVID-19 vaccine acceptance [44–46]. People with high preventive practices related to mpox also demonstrate significantly higher vaccination willingness implies that individuals who are more engaged in health-protective behaviors are more likely to trust and accept vaccines, reinforcing the link between preventive actions and vaccine confidence [47].

Interestingly, the study finds a small but positive impact of education, reinforcing the notion that higher education often correlates with greater health literacy and vaccine acceptance, which has been observed during COVID-19 pandemics [48]. Regarding urban-rural disparities in vaccine acceptance in China, a large-scale national study on COVID-19 vaccine acceptance found that urban participants were more hesitant to receive the vaccine compared to their rural counterparts [49]. In contrast, this study shows that urban residents exhibited a higher willingness to vaccinate against mpox, emphasizing the potential role of health literacy [50]. Concerning income disparities in vaccine acceptance, several studies in China have found that higher income levels were linked to COVID-19 vaccine hesitancy [51]. This was rationalized by the fact that individuals with higher incomes have greater access to vaccine information through social media, including misinformation, which can heighten concerns about vaccine safety and efficacy [52]. Similarly, an inverse relationship between income and mpox vaccine acceptance was observed in this study, although the association was not statistically significant. This may reflect the more complex socio-economic dynamics surrounding mpox, where fear may outweigh financial considerations. Further study is needed to confirm this finding. Lastly, the lack of impact of gender, age, and attitude towards EUA vaccines suggests that these factors may play a lesser role in the context of mpox.

The descriptive findings of vaccine preferences in this study reveal important insights into public sentiment towards different vaccine types in the context of China. The findings show that a larger segment of the population favors traditional vaccines for mpox. This trend aligns with the broader context of China's vaccination landscape, where traditional vaccines, particularly inactivated virus vaccines like Sinovac and Sinopharm, were extensively used during the COVID-19 pandemic. A multi-country study that includes China found that the novelty of the mRNA vaccine for COVID-19 increases hesitancy, with the single-country study in China revealing a preference of 53.2% for conventional vaccines compared to 50.9% for mRNA vaccines among participants [52]. The established safety record and familiarity of these conventional vaccines, widely used during the COVID-19 pandemic in China, likely contribute to the higher public confidence and acceptance of traditional vaccines. Notably, the first Chinese mpox vaccine was approved for clinical trials on September 9, 2024. Developed independently by the Shanghai Institute of Biological Products, a subsidiary of China National Pharmaceutical Group (Sinopharm), this vaccine is based on a live, attenuated orthopoxvirus [53]. Our findings, showing a preference for conventional vaccines, are timely and offer valuable insights into the potential acceptance of this vaccine once it becomes available.

Lastly, the results of the WTP for a potential mpox vaccine revealed a relatively lower amount that participants were willing to pay, highlighting potential barriers to accessing the mpox vaccine should it become available at a cost. Given China's diverse socio-economic landscape, where disparities in income levels exist, pricing strategies for future vaccines must be carefully considered to ensure equitable access across different demographic groups. However, it is important to note that the demographic characteristics of the study participants reflect a predominantly young, educated, and urban population with relatively high-income levels, which may limit the generalizability of the findings to broader populations. In summary, the findings emphasize the need for public health initiatives in China to focus not only on promoting the safety and efficacy of vaccines but also on addressing affordability concerns to enhance vaccine acceptance and uptake in the context of emerging infectious diseases like mpox.

**Limitations**

While this study has several strengths, such as its large sample size, rigorous data collection methods, and broad representation across major provinces in China, it also has some limitations. One notable limitation is the potential bias stemming from the non-random selection of participants in the online survey, which may not fully capture the diversity of the entire population. These biases could limit the generalizability of the findings to the broader population of China. Additionally, the reliance on self-reported data introduces the possibility of recall bias and social desirability bias. Furthermore, as a cross-sectional study, it is unable to establish causal relationships between variables.

Of note, despite the intended target of 768 participants per region, actual sample sizes varied, with the Northwestern and Southwestern regions notably underrepresented (77 and 126 respondents, respectively). The imbalance in participation may be influenced by regional socioeconomic disparities in China. The Eastern and Southern Central regions, which have relatively large sample sizes, are more economically developed, urbanized, and have higher levels of education and digital engagement. These factors likely contribute to greater awareness of health-related issues and a higher willingness to participate in online surveys. This uneven distribution may limit the generalizability of the findings, as smaller sample sizes reduce statistical power and increase the risk of bias in regional comparisons. Underrepresentation of certain areas may also skew results, as perspectives from these regions may not be fully captured, potentially affecting conclusions on vaccination willingness. Moreover, the imbalance in sample distribution may lead to a demographic profile that is disproportionately young, educated, urban, and higher-income. To improve representativeness and reliability, future studies should adopt targeted recruitment strategies and ensure a more balanced sample across all regions.

Furthermore, it is also important to note that the decision to categorize responses by region rather than by province limits the ability to analyze provincial-level variations, which may have important consequences. Given the diverse socio-economic, cultural, and healthcare landscapes across China's provinces, aggregating data at the regional level may conceal significant differences in vaccine confidence. This limitation affects the applicability of findings for policymakers, as province-specific insights are crucial for tailoring interventions.

Based on the findings of this study, specific recommendations to enhance vaccine confidence and acceptance include implementing comprehensive educational campaigns to increase awareness of mpox symptoms and risks, particularly in regions with lower knowledge levels. Strengthening public health messaging, even in the absence of reported cases, can help sustain preventive practices and preparedness. Efforts should also focus on addressing concerns regarding EUA vaccines by transparently communicating their rigorous approval processes and safety measures. Tailored outreach strategies should be developed to address regional disparities in vaccine acceptance, particularly in economically developed areas where hesitancy may be higher. Additionally, affordability concerns must be considered to ensure equitable access to vaccines across diverse socio-economic groups. Given the strong preference for traditional vaccines, building confidence in mRNA and other new vaccine technologies requires emphasizing their rigorous safety evaluations, highlighting successful real-world applications, and ensuring clear, transparent communication about their benefits and potential risks. Lastly, we acknowledge that this study was conducted during a period without reported mpox cases and with limited

media coverage in China, which may have influenced public awareness, perceptions, and willingness to receive the mpox vaccine.

## Conclusion

In conclusion, the findings highlight the complex interplay between regional economic disparities, socio-economic factors, knowledge, and attitudes toward vaccine acceptance in China. While less economically developed regions, such as the Northeast and Northwest, showed higher willingness to receive an EUA-authorized mpox, wealthier regions like the East and North exhibited lower or moderate willingness, challenging the assumption that higher economic development correlates with greater vaccine uptake. Trust in future EUA vaccines, shaped by lessons learned from the COVID-19 vaccination experience, emerged as the most significant factor influencing willingness. This suggests that public confidence in the safety and efficacy of emergency-authorized vaccines plays a pivotal role in shaping vaccine acceptance, especially during health crises. Although participants exhibited relatively low-risk perceptions, this factor still played a significant role in shaping vaccine willingness, highlighting the importance of addressing both trust and risk awareness in public health strategies. Additionally, preventive practices related to mpox also significantly shape vaccination willingness highlighting the importance of awareness and proactive actions in promoting vaccine acceptance. By demographics, while education also played a significant positive role, factors such as income, gender, and age did not significantly impact mpox vaccine acceptance. These results underscore the need for targeted public health interventions and tailored communication strategies to address regional disparities and enhance future mpox vaccine confidence.

## Supporting information

**S1 Appendix. Survey questionnaire.**
(DOCX)

**S2 Appendix. Proportion of correct responses for knowledge of mpox symptoms.**
(TIF)

**S3 Appendix. Risk perception towards mpox.**
(TIF)

**S4 Appendix. Proportion of responses for mpox prevention practices.**
(TIF)

**S5 Appendix. Proportion of responses for attitudes towards EUA vaccines.**
(TIF)

## Acknowledgments

We would like to thank the study participants for participating in this study.

## Author contributions

**Conceptualization:** Li Ping Wong, Hai Yen Lee, Yulan Lin.

**Data curation:** Zhiwen Huang, Yulan Lin.

**Formal analysis:** Li Ping Wong, Haridah Alias.

**Funding acquisition:** Yulan Lin.

**Investigation:** Li Ping Wong, Zhijian Hu, Yulan Lin.

**Methodology:** Li Ping Wong, Qinjian Zhao, Zhijian Hu, Yulan Lin.

**Project administration:** Yulan Lin.

**Resources:** Yulan Lin.

**Software:** Li Ping Wong, Haridah Alias.

**Supervision:** Li Ping Wong, Hai Yen Lee, Zhijian Hu, Yulan Lin.

**Validation:** Li Ping Wong, Yulan Lin.

**Visualization:** Li Ping Wong, Haridah Alias.

**Writing – original draft:** Li Ping Wong.

**Writing – review & editing:** Li Ping Wong, Haridah Alias, Hai Yen Lee, Qinjian Zhao, Zhiwen Huang, Zhijian Hu, Yulan Lin.

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
