## [Decision Letter · Decision Letter 0]

18 Feb 2025

PNTD-D-24-01701

Investigating post-COVID-19 Confidence in Emergency Use Authorization Vaccines: A Hypothetical Case of Monkeypox

Dear Dr. Lin,

Thank you for submitting your manuscript to PLOS Neglected Tropical Diseases. After careful consideration, we feel that it has merit but does not fully meet PLOS Neglected Tropical Diseases's publication criteria as it currently stands. Therefore, we invite you to submit a revised version of the manuscript that addresses the points raised during the review process.

Please submit your revised manuscript within 60 days Apr 19 2025 11:59PM. If you will need more time than this to complete your revisions, please reply to this message or contact the journal office at plosntds@plos.org. Please include the following items when submitting your revised manuscript:

We look forward to receiving your revised manuscript.

Kind regards,

Richard A. Bowen, DVM PhD

Academic Editor

Abdallah Samy

Section Editor

Shaden Kamhawi

co-Editor-in-Chief

Paul Brindley

co-Editor-in-Chief

**Additional Editor Comments:**

Your manuscript was well received and all reviewers provided comments and suggestions that should further increase the value of your manuscript. Please evaluate reviewer comments, modifiy your manuscript accordingly and provide a response to all reviewer comments. We look forward to receiving a revision of this valuable contribution.

**Journal Requirements:**

At this stage, the following Authors/Authors require contributions: Li Ping Wong, Haridah Alias, Hai Yen Lee, Qinjian Zhao, Zhiwen Huang, Zhijian Hu, and Yulan Lin. Please ensure that the full contributions of each author are acknowledged in the "Add/Edit/Remove Authors" section of our submission form.

3) Tables should not be uploaded as individual files. Please remove these files and include the Tables in your manuscript file as editable, cell-based objects. For more information about how to format tables, see our guidelines:

https://journals.plos.org/plosntds/s/tables

Potential Copyright Issues:

- Figure 1. Please provide a direct link to the base layer of the map (i.e., the country or region border shape) and ensure this is also included in the figure legend; and provide a link to the terms of use / license information for the base layer image or shapefile. We cannot publish proprietary or copyrighted maps (e.g. Google Maps, Mapquest) and the terms of use for your map base layer must be compatible with our CC BY 4.0 license.

**Reviewers' Comments:**

Reviewer's Responses to Questions

**Key Review Criteria Required for Acceptance?**

**Methods** :

-Are the objectives of the study clearly articulated with a clear testable hypothesis stated?

-Is the study design appropriate to address the stated objectives?

-Is the population clearly described and appropriate for the hypothesis being tested?

-Is the sample size sufficient to ensure adequate power to address the hypothesis being tested?

-Were correct statistical analysis used to support conclusions?

-Are there concerns about ethical or regulatory requirements being met?

Reviewer #1: Major comments

1. Study authors note that they used convivence sampling, but fail to mention how this sample was generally recruited. How was the link to the survey sent to people, etc. that may bias this sample? Additionally information may help determine if the crude assumption of a design effect of 2 is adequate.

2. How many provinces were included? Given the goal sample size of over 750 per province, the final sample size for the survey seems quite small. Additional information is required. I presume that this survey was powered to detect difference in response by province, though this was never explicitly stated. Then, there was no mention of province again following the beginning of the methods; this is concerning. Even by region, which perhaps contains provinces, there was not enough survey responses collected. This needs to be clarified.

3. Given the lack of mention of human-to-human transmission in the paper’s introduction, what did study authors explicitly mean with respect to “avoiding close contact, practicing good hygiene, avoiding face-touching, using personal protective equipment (PPE), and staying informed” in their survey questionnaire? What was the precise language used and did it use language suggesting just animal-to-human or both animal-to-human and human-to-human transmission possibilities?

4. The introduction mentions development of new vaccines, but fails to mention that an already existing JYNNEOS vaccine is available for high-risk persons. First, this should be mentioned somewhere in the paper as study authors leave out this possibility all together. Second, was this considered in the survey ?

Minor comments

1. Line 210-212: Please correct missing ethics statement.

2. What software was used for statistical analyses, and is the code publicly accessible?

3. Table 1 could contain much more comprehensive information, including key survey question results broken down by these demographic characteristics.

Reviewer #2: Overall, the study objectives are clearly articulated, and the nationwide cross-sectional design effectively provides a snapshot of mpox knowledge and willingness to receive the mpox vaccine.

1) However, it is not clear how the convenient sampling has been conducted. It is stated a company conducted random sampling across six regions in China. Could you elaborate on the targeted population and how participants were recruited for the survey? Specifically, how did they receive the survey link, and what methods were used to access the survey (e.g., email, social media, etc.)?

2) In lines 457-458, you mention 'One notable limitation is the potential bias stemming from the non-random selection of participants in the online survey, which may not fully capture the diversity of the entire population.' However, this contradicts the methods section, where random selection is described. Please clarify this discrepancy for the reader.

3) In lines 135-137, you state, 'The sample size was multiplied by the predicted design effect of two to account for the use of convenience sampling and an online survey. Hence, the optimal sample size was set to 768 (384 x 2).' However, with six regions, the total optimal sample size should be 4608 (768 per region). The overall sample size was 2000. Does this discrepancy reflect a different sampling approach or is there another explanation? Additionally, could you specify how many surveys were excluded due to incompleteness?

4) Please include a statement regarding whether participants were compensated for their participation in the study, as part of the ethical approval section.

Reviewer #3: The methods section is well detailed with study objectives clearly explained. The section could use a little more detail on anticipated biases for the sampling method used. The sample size seems okay and the calculation is well detailed. Statistical analyses were ably explained

Reviewer #4: - Objectives of the study are clearly articulated with well thought-out analyses. There was no hypothesis given the study's nature (exploratory).

- The study design is appropriate to address the research question and the population size is sufficient for representation.

- The correct statistical analyses were used and there are no concerns with ethics as it was a cross-sectional study design.

- Please clarify if the sampling method was convenience or randomized as the way the statement is written is confusing (the limitation section mentions that the sample is non-random).

A few points regarding methods:

- the paragraph "Knowledge about mpox symptoms" (line 147) paragraph should be written in past tense

- check for uniform capitalization for variables and options (eg. choose whether you want "Never" or "never" and keep it consistent throughout the methods)

- remove lines 241-243

- minor grammar errors that need to be fixed

**Results** :

-Does the analysis presented match the analysis plan?

-Are the results clearly and completely presented?

-Are the figures (Tables, Images) of sufficient quality for clarity?

Reviewer #1: Please see comments above; results may be inadequate if methods concerns are not addressed.

Reviewer #2: The results section presents a comprehensive volume of data directly related to the research question. Additionally, it is beneficial that the characteristics of the respondents are clearly outlined at the beginning of this section. Overall, the results were conclusive and well-presented.

1) When mentioning CNY, please also include the equivalent amount in USD for clarity, especially for readers who may not be familiar with the currency exchange rates.

2) Figures/Tables: Please provide higher-resolution figures and tables to enhance readability (e.g. Appendix 2-5).

3) Please double-check Figure 2.

- The value for preventive practice is stated as 0.066 in the text but is shown as 0.064 in the figure. Additionally, the values for knowledge and susceptibility do not match those presented in the text (lines 293-296).

- Line 297: Is 'location' referring to 'birthplace'? This is unclear in Figure 2 and should be clarified.

Reviewer #3: The results section is detailed but text heavy. The two figures and one table at the end of the references section are not sufficient to provide enough visual presentation of the results

Reviewer #4: The analyses were well executed and matched the analysis plan.

Issues to fix:

- Methods stated that ethnicity was captured but it is not in Table 1 of results or the first paragraph of results. Was there a reason as to why ethnicity was not mentioned? If not, please add that stat to the table and results section.

- Line 238, please re-word the sentence (minor grammar error)

- Please contextualize the monthly income (high vs middle vs low/below poverty line) for researchers outside of the country

- The images in appendix are hard to read, please ensure that they are readable in the full publication

- For the "descriptive analysis of vaccine preference" (lines 313-314), the percentages add up to 85% and not 100%. Please revise or clarify of why that is the case.

**Conclusions** :

-Are the conclusions supported by the data presented?

-Are the limitations of analysis clearly described?

-Do the authors discuss how these data can be helpful to advance our understanding of the topic under study?

-Is public health relevance addressed?

Reviewer #1: Please see comments above; conclusions may be inadequate if methods concerns are not addressed.

Reviewer #2: The conclusion is well-supported by the data and clearly connects to the research question. However, clarifying the factors behind regional variations in vaccine acceptance could strengthen it. Exploring whether regions with higher willingness for the mpox vaccine show similar trends with other vaccines or campaigns would be valuable, as would discussing strategies to increase trust in the target population.

The limitations are addressed, but providing more detail on the varying sample sizes across regions would improve transparency.

The authors effectively explain how the data advance understanding of vaccine acceptance. While the public health relevance is clear, including specific recommendations for improving vaccine confidence would enhance the practical implications of the study.

Reviewer #3: Conclusions and limitations are valid and relevant at detailing pubic health relevance of the study

Reviewer #4: The discussion is a bit lacking. There are multiple instances of claims being made with no evidence (eg. line 339 - 340, 353-355, 367-369, etc), please add citations for any claims made.

- please fix some minor grammar errors

**Editorial and Data Presentation Modifications?**

Reviewer #1: (No Response)

Reviewer #2: (No Response)

Reviewer #3: Can be accepted pending "Minor Revision" and comments in the copy of the manuscript I have uploaded.

Reviewer #4: Current data is sufficient, please ensure that figures are clear upon publication (currently they show up very pixelated and hard to read). I recommend this manuscript for "Minor Revision"

**Summary and General Comments** :

Reviewer #1: Thank you for the opportunity to review this manuscript. Study authors present a survey on knowledge of mpox and acceptability of a possible future vaccine EUA. This topic is interesting and important as mpox continues to spread globally. However, I have significant concerns regarding the survey design and communication of such design.

Reviewer #2: This study addresses a timely issue by examining post-COVID-19 confidence in EUA vaccines using a hypothetical mpox case. The manuscript is well-structured and offers valuable insights, but refining the methodology for clarity, specifically the process of convenience sampling and clarifying the sample size, would strengthen the conclusions. It's also important to note that this study was conducted in the absence of mpox cases and with limited media coverage in China.

The limited media coverage and absence of mpox cases may have impacted participants' perceptions and vaccine willingness, potentially limiting the generalizability of the findings. These factors are appropriately acknowledged in the limitations and discussion sections.

Additionally, the term 'mpox' should be used consistently throughout the manuscript (e.g., in the title) to help reduce stigma. Overall, this is a relevant and well-conceived study that deserves publication.

Specific comments:

Title: I would suggest using the term 'mpox' instead of 'monkeypox' to reduce stigma and avoid discrimination.

Abstract:

- Conclusion: Consider adding strategies to strengthen public trust in vaccines within your target population.

Introduction:

- Line 119: Please define 'NTDs' upon first use to ensure clarity for all readers

Results:

- Line 227: 'A total of 2000 participants completed responses were received and included in the analysis.'

I suggest adding the number of surveys excluded due to incompleteness for clarity.

- Line 262-264: 'Many participants also worried about the effectiveness of EUA vaccines compared to fully approved ones (46.7% agreed, 29.5% strongly agreed) and potential unknown long-term side effects (48.8% agreed, 27.6% strongly agreed).'

This is an interesting finding. Is there an explanation for why the effectiveness of EUA vaccines is perceived as lower? This could be a valuable addition to the discussion section.

Discussion:

- Line 349-352: 'The results on attitudes toward EUA vaccines revealed that many participants expressed concerns about the rapid approval process, quality control, and doubts about their effectiveness compared to fully approved vaccines.'

This is an interesting topic. Are there other publications that discuss why an EUA vaccine may be perceived as less effective?

- Lines 365-366: I would suggest adding information about the guidelines and Standard Operating Procedures (SOPs) for EUA vaccines in China to provide more context.

- Lines 371-379: This is an interesting statement. What was the rationale behind China’s decision not to use mRNA vaccines for COVID-19? Is there more information on the reasoning behind this choice?

- Lines 287-288 and 382-385: 'These findings indicate regional variation in vaccine acceptance, with higher rates in the northeastern and northwestern areas.'

Are there any previous studies or known factors that explain why these regions may have higher vaccine acceptance? Has this trend been observed in other vaccination campaigns or governmental initiatives in those regions?

- Lines 390-392: 'Tailored communication strategies and outreach efforts could be vital in boosting vaccine confidence, particularly in regions with low acceptance.'

What specific strategies could be implemented? Do similar measures already exist from other campaigns?

- Lines 392-394: Please delete the duplicated sentence for clarity.

Limitation:

- Line 464-466: 'For example, while the Eastern and Southern Central regions have relatively large sample sizes, other regions, such as the Northwestern and Southwestern, have significantly fewer participants.'

What factors contributed to this imbalance in participation?

Reviewer #3: This is great write up on the topic, worth acceptance pending minor revisions

Reviewer #4: Thank you, it was a pleasure to review your publication. This manuscript is well done and thought out; it just needs a few edits and some points of clarification. I think this will add to the body of literature that aids to fight misinformation and fear surrounding vaccines on a global level. While the study had a simple, cross-sectional design, the analyses were well done. Furthermore, the study reported crucial data regarding attitudes, knowledge, and risk perception surrounding Mpox and vaccines in general. I recommend this manuscript for "Minor Revision."

PLOS authors have the option to publish the peer review history of their article (what does this mean? ). If published, this will include your full peer review and any attached files.

**Do you want your identity to be public for this peer review?** For information about this choice, including consent withdrawal, please see our Privacy Policy .

Reviewer #1: No

Reviewer #2: **Yes: ** Lena Huebl

Reviewer #3: **Yes: ** Adamson Paxon NDHLOVU

Reviewer #4: No

**Figure resubmission:**
---

## [Editor Report · Decision Letter 1]

4 Apr 2025

Dear Dr. Lin,

We are pleased to inform you that your manuscript 'Investigating post-COVID-19 Confidence in Emergency Use Authorization Vaccines: A Hypothetical Case of Mpox' has been provisionally accepted for publication in PLOS Neglected Tropical Diseases.

Best regards,

Richard A. Bowen, DVM PhD

Academic Editor

Abdallah Samy

Section Editor

Shaden Kamhawi

co-Editor-in-Chief

Paul Brindley

co-Editor-in-Chief

---

## [Editor Report · Acceptance letter]

Dear Dr. Lin,

We are delighted to inform you that your manuscript, "Investigating post-COVID-19 Confidence in Emergency Use Authorization Vaccines: A Hypothetical Case of Mpox," has been formally accepted for publication in PLOS Neglected Tropical Diseases.

Best regards,

Shaden Kamhawi

co-Editor-in-Chief

Paul Brindley

co-Editor-in-Chief
